# SMALL CORESETS TO REPRESENT LARGE TRAINING DATA FOR SUPPORT VECTOR MACHINES

## ABSTRACT

Despite their popularity, even efficient implementations of Support Vector Machines (SVMs) have proven to be computationally expensive to train at a large-scale, especially in streaming settings. In this paper, we propose a coreset construction algorithm for efficiently generating compact representations of massive data sets to speed up SVM training. A coreset is a weighted subset of the original data points such that SVMs trained on the coreset are provably competitive with those trained on the original (massive) data set. We provide lower and upper bounds on the number of samples required to obtain accurate approximations to the SVM problem as a function of the complexity of the input data. Our analysis also establishes sufficient conditions on the existence of sufficiently compact and representative coresets for the SVM problem. We empirically evaluate the practical effectiveness of our algorithm against synthetic and real-world data sets.

## 1 INTRODUCTION

Popular machine learning algorithms are computationally expensive, or worse yet, intractable to train on Big Data. The notion of using *coresets* (Feldman & Langberg, 2011; Braverman et al., 2016; Bachem et al., 2017), small weighted subsets of the input points that provably approximate the original data set, has shown promise in accelerating machine learning algorithms such as $k$-means clustering (Feldman & Langberg, 2011), mixture model training (Feldman et al., 2011; Lucic et al., 2017), and logistic regression (Huggins et al., 2016).

Coreset constructions were originally introduced in the context of computational geometry (Agarwal et al., 2005) and subsequently generalized for applications to other problems (Langberg & Schulman, 2010; Feldman & Langberg, 2011). Coresets provide a compact representation of the structure of static and streaming data, with provable approximation guarantees with respect to specific algorithms. For instance, a data set consisting of $K$ clusters would yield a coreset of size $K$, with each cluster represented by one coreset point. Even if the data has no structure (e.g., uniformly distributed), coresets will correctly down sample the data to within prescribed error bounds. For domains where the data has structure, the coreset representation has the potential to greatly and effectively reduce the time required to manually label data for training and the computation time for training, while at the same time providing a mechanism of supporting machine learning systems for applications with streaming data.

Coresets are constructed by approximating the relative importance of each data point in the original data set to define a sampling distribution and sampling sufficiently many points in accordance with this distribution. This construction scheme suggests that beyond providing a means of conducting provably fast and accurate inference, coresets also serve as efficient representations of the full data set and may be used to automate laborious representation tasks, such as automatically generating semantic video representations or detecting outliers in data (Lucic et al., 2016).

The representative power and provable guarantees provided by coresets also motivate their use in training of one of the most popular algorithms for classification and regression analysis: Support Vector Machines (SVMs). Despite their popularity, SVMs are computationally expensive to train on massive data sets, which has proven to be computationally problematic with the rising availability of Big Data. In this paper, we present a novel coreset construction algorithm for efficient, large-scale Support Vector Machine training.

In particular, this paper contributes the following:

1. A practical coreset construction algorithm for accelerating SVM training based on an efficient importance evaluation scheme for approximating the importance of each point.

2. An analysis proving lower bounds on the number of samples required by any coreset construction algorithm to approximately represent the data.

3. An analysis proving the efficiency and theoretical guarantees of our algorithm and characterizing the family of data sets for which applications of coresets are most suited.

4. Evaluations against synthetic and real-world data sets that demonstrate the practical effectiveness of our algorithm for large-scale SVM training.

## 2 RELATED WORK

Training a canonical Support Vector Machine (SVM) requires $\mathcal{O}(n^3)$ time and $\mathcal{O}(n^2)$ space where $n$ is the number of training points (Tsang et al., 2005). Work by Tsang et al. (2005) introduced Core Vector Machines (CVMs) that reformulated the SVM problem as the Minimum Enclosing Ball (MEB) problem and used existing coreset methods for MEB to compress the data. The authors proposed a method that generates a $(1 + \varepsilon)^2$ approximation to the two-class L2-SVM in $\mathcal{O}(n/\varepsilon^2 + 1/\varepsilon^4)$ time, when certain assumptions about the kernel used are satisfied. However, CVM's accuracy and convergence properties have been noted to be at times inferior to the performance of existing SVM implementations (Loosli & Canu, 2007). Similar geometric approaches including extensions to the MEB formulation, those based on convex hulls, and extreme points, among others, were also investigated by Rai et al. (2009); Agarwal & Sharathkumar (2010); Har-Peled et al. (2007); Nandan et al. (2014).

Since the SVM problem is inherently a quadratic optimization problem, prior work has investigated approximations to the quadratic programming problem using the Frank-Wolfe algorithm or Gilbert's algorithm (Clarkson, 2010). Another line of research has been in reducing the problem of polytope distance to solve the SVM problem (Gärtner & Jaggi, 2009). The authors establish lower and upper bounds for the polytope distance problem and use Gilbert's algorithm to train an SVM in linear time.

A variety of prior approaches were based on randomized algorithms with the property that they generated accurate approximations with high probability. Most notable are the works of Clarkson et al. (2012) Hazan et al. (2011). Hazan et al. (2011) used a primal-dual approach combined with Stochastic Gradient Descent (SGD) in order to train linear SVMs in sub-linear time. They proposed the SVM-SIMBA approach and proved that it generates an $\varepsilon$-approximate solution with probability at least $1/2$ to the SVM problem that uses hinge loss as the objective function. The key idea in their method is to access single features of the training vectors rather than the entire vectors themselves. Their method is nondeterministic and returns the correct $\varepsilon$-approximation with probability greater than a constant probability, similar to the probabilistic guarantees of coresets.

Clarkson et al. (2012) present sub-linear-time (in the size of the input) approximation algorithms for some optimization problems such as training linear classifiers (e.g., perceptron) and finding MEB. They introduce a technique that is originally applied to the perceptron algorithm, but extend it to the related problems of MEB and SVM in the hard margin or $L2$-SVM formulations. Shalev-Shwartz et al. (2011) introduce Pegasos, a stochastic sub-gradient algorithm for approximately solving the SVM optimization problem, that runs in $\widetilde{\mathcal{O}}(dnC/\varepsilon)$ time for a linear kernel, where $C$ is the SVM regularization parameter and $d$ is the dimensionality of the input data points. These works offer probabilistic guarantees, similar to those provided by coresets, and have been noted to perform well empirically; however, unlike coresets, SGD-based approaches cannot be trivially extended to streaming settings since each new arriving data point in the stream results in a change of the gradient.

Joachims (2006) presents an alternative approach to training SVMs in linear time based on the cutting plane method that hinges on an alternative formulation of the SVM optimization problem. He shows that the Cutting-Plane algorithm can be leveraged to train SVMs in $\mathcal{O}(sn)$ time for classification and $\mathcal{O}(sn \log n)$ time for ordinal regression where $s$ is the average number of non-zero features. Har-Peled et al. (2007) constructs coresets to approximate the maximum margin separation, i.e., a hyperplane that separates all of the input data with margin larger than $(1-\varepsilon)\rho^*$, where $\rho^*$ is the best achievable margin.

## 3 PROBLEM DEFINITION

### 3.1 SOFT-MARGIN SVM

We assume that we are given a set of weighted training points $\mathcal{P} = \{(x_i, y_i)\}_{i=1}^n$ with the corresponding weight function $u : \mathcal{P} \to \mathbb{R}_{\geq 0}$, such that for every $i \in [n]$, $x_i \in \mathbb{R}^d$, $y_i \in \{-1, 1\}$, and $u(p_i)$ corresponds to the weight of point $p_i$. For simplicity, we assume that the bias term is embedded into the feature space by defining $\tilde{x}_i = (x_i, 1)$ for each point and $\tilde{w} = (w, 1)$ for each query. Thus, we henceforth assume that we are dealing with $d+1$ dimensional points and refer to $\tilde{w}$ and $\tilde{x}_i$ as just $w$ and $x_i$ respectively. Under this setting, the hinge loss of a point $p_i = (x_i, y_i)$ with respect to the separating hyperplane $w$ is defined as $h(p_i, w) = [1 - y_i \langle x_i, w \rangle]_+$, where $[x]_+ = \max\{0, x\}$.

For any subset of points, $\mathcal{P}' \subseteq \mathcal{P}$, define $H(\mathcal{P}', w) = \sum_{p \in \mathcal{P}'} u(p) h(p, w)$ as the sum of the hinge losses and $u(\mathcal{P}') = \sum_{p \in \mathcal{P}'} u(p)$ as the sum of the weights of points in set $\mathcal{P}'$. To clearly depict the contribution of each point to the objective value of the SVM problem, we present the SVM objective function, $f(\mathcal{P}, w)$, as the sum of per-point objective function evaluations, which we formally define below.

**Definition 1** (Per-point Objective Function). *Define the function* $f : (\mathbb{R}^{d+1} \times \{-1, 1\}) \times \mathbb{R}^{d+1} \to \mathbb{R}_{\geq 0}$ *such that for every* $i \in [n]$

$$f(p_i, w) = \frac{\|w_{1:d}\|_2^2}{2u(\mathcal{P})} + Ch(p_i, w),$$

$w_{1:d}$ *denotes the entries* $1 : d$ *of* $w$, $h(p_i, w)$ *is the corresponding hinge loss, and* $C \in [0, 1]$ *is the regularization parameter.*

**Definition 2** (Soft-margin SVM Problem). *Given a set of* $d + 1$ *dimensional weighted points* $\mathcal{P}$ *with weight function* $u : \mathcal{P} \to \mathbb{R}_{\geq 0}$ *the primal of the SVM problem is expressed by the following quadratic program*

$$\min_{w \in \mathcal{Q}} f((\mathcal{P}, u), w), \tag{1}$$

*where* $f$ *is the evaluation of the weighted point set* $\mathcal{P}$ *with weight function* $u : \mathcal{P} \to \mathbb{R}_{\geq 0}$,

$$f((\mathcal{P}, u), w) = \sum_{i \in [n]} u(p_i) f(p_i, w) = \frac{\|w_{1:d}\|_2^2}{2} + CH(\mathcal{P}, w). \tag{2}$$

When the set of points $\mathcal{P}$ and the corresponding weight function $u$ are clear from context, we will henceforth denote $f((\mathcal{P}, u), w)$ by $f(\mathcal{P}, w)$ for notational convenience.

### 3.2 CORESETS

Coresets can be seen as a compact representation of the full data set that approximate the SVM cost function (2) uniformly over all queries $w \in \mathcal{Q}$. Thus, rather than introducing an entirely new algorithm for solving the SVM problem, our approach is to reduce the runtime of standard SVM algorithms by compressing the size of the input points from $n$ to a compact set whose size is sublinear (ideally, polylogarithmic) in $n$.

**Definition 3** ($\varepsilon$-coreset). *Let* $\varepsilon \in (0, 1/2)$ *and* $\mathcal{P} \subset \mathbb{R}^{d+1} \times \{-1, 1\}$ *be a set of* $n$ *weighted points with weight function* $u : \mathcal{P} \to \mathbb{R}_{\geq 0}$. *The weighted subset* $(\mathcal{S}, v)$, *where* $\mathcal{S} \subset \mathcal{P}$ *with corresponding weight function* $v : \mathcal{S} \to \mathbb{R}_{\geq 0}$ *is an* $\varepsilon$-coreset *if for any query* $w \in \mathcal{Q}$, $(\mathcal{S}, v)$ *satisfies the coreset-property*

$$|f((\mathcal{P}, u), w) - f((\mathcal{S}, v), w)| \leq \varepsilon f((\mathcal{P}, u), w) \quad \textit{(Coreset Property)}. \tag{3}$$

Our overarching goal is to efficiently construct an $\varepsilon$-coreset, $(\mathcal{S}, v)$, such that the size of $\mathcal{S}$ is sufficient small in comparison to the original number of points $n$.

## 4 METHOD

### 4.1 METHOD OVERVIEW

Our coreset construction scheme is based on the unified framework of Langberg & Schulman (2010); Feldman & Langberg (2011) and is shown as Alg. 1. The crux of our algorithm lies in generating the importance sampling distribution via efficiently computable upper bounds (proved in Sec. 5) on the importance of each point (Lines 1-6). Sufficiently many points are then sampled from this distribution and each point is given a weight that is inversely proportional to its sample probability (Lines 7-9). The number of points required to generate an $\varepsilon$-coreset with probability at least $1 - \delta$ is a function of the desired accuracy $\varepsilon$, failure probability $\delta$, and complexity of the data set ($t$ from Theorem 9). Under mild assumptions on the problem at hand (see Sec. 5.2), the required sample size is polylogarithmic in $n$.

Intuitively, our algorithm can be seen as an importance sampling procedure that first generates a judicious sampling distribution based on the structure of the input points and samples sufficiently many points from the original data set. The resulting weighted set of points, $(S, v)$, serves as an unbiased estimator for $f(\mathcal{P}, w)$ for any query $w \in \mathcal{Q}$, i.e., $\mathbb{E}[f((S, v), w)] = f(\mathcal{P}, w)$. Although sampling points uniformly with appropriate weights can also generate such an unbiased estimator, it turns out that the variance of this estimation is minimized if the points are sampled according to the distribution defined by the ratio between each point's sensitivity and the sum of sensitivities, i.e., $\gamma(p_i)/t$ on Line 9 (Bachem et al., 2017).

### 4.2 CHICKEN AND THE EGG PHENOMENA

Coresets are intended to provide efficient and provable approximations to the optimal SVM solution, however, the very first line of our algorithm entails computing the optimal solution to the SVM problem. This seemingly eerie phenomenon is explained by the merge-and-reduce technique (Har-Peled & Mazumdar, 2004) that ensures that our coreset algorithm is only run against small partitions of the original data set (Har-Peled & Mazumdar, 2004; Braverman et al., 2016; Lucic et al., 2017). The merge-and-reduce approach leverages the fact that coresets are composable and reduces the coreset construction problem for a (large) set of $n$ points into the problem of computing coresets for $\frac{n}{2|S|}$ points, where $2|S|$ is the minimum size of input set that can be reduced to half using Alg. 1 (Braverman et al., 2016). Assuming that the sufficient conditions for obtaining polylogarithmic size coresets implied by Theorem 9 hold, the overall time required for coreset construction is nearly linear in $n$, $\widetilde{\mathcal{O}}_{\varepsilon,\delta}(d^3 n)$[1]. This follows from the fact that $2|S| = \widetilde{\mathcal{O}}_{\delta,\varepsilon}(d)$ by Theorem 9, that the Interior Point Method runs in time $\mathcal{O}(|S|^3) = \tilde{\mathcal{O}}_{\delta,\varepsilon}(d^3)$ for an input set of size $2|S|$, and that the merge-and-reduce tree has height at most $\lceil \log n \rceil$, meaning that an accuracy parameter of $\varepsilon' = \varepsilon/\log n$ has to be used in the intermediate coreset constructions to account for the compounded error over all levels of the tree (Braverman et al., 2016).

### 4.3 EXTENSIONS

We briefly remark on a straightforward extension that can be made to our algorithm to accelerate performance and applicability. In particular, the computation of the optimal solution to the SVM problem in line 1 can be replaced by an efficient gradient-based method, such as Pegasos (Shalev-Shwartz et al., 2011), to compute an approximately $\xi$ optimal solution in $\widetilde{\mathcal{O}}(dnC/\xi)$ time, which is particularly suited to scenarios with $C$ small. We give this result as Lemma 11, an extension of Lemma 7. We also note that based on our analytical results (Lemmas 7 and 11), any SVM solver, either exact or approximate, can be used in Line 1 as a replacement for the Interior Point Method.

## 5 ANALYSIS

In this section, we prove upper and lower bounds on the sensitivity of a point in terms of the complexity of the given data set. Our main result is Theorem 9, which establishes sufficient conditions

---

[1] $\widetilde{\mathcal{O}}_{\varepsilon,\delta}$ notation suppresses $\varepsilon, \delta$ and polylog($n$) factors.

---

**Algorithm 1:** CORESET($\mathcal{P}, u, \varepsilon, \delta$)

---

**Input:**      A set of training points $\mathcal{P} \subseteq \mathbb{R}^{d+1} \times \{-1, 1\}$ containing $n$ points,
               a weight function $u : \mathcal{P} \to \mathbb{R}_{\geq 0}$,
               an error parameter $\varepsilon \in (0, 1)$, and failure probability $\delta \in (0, 1)$.

**Output:**     An $\varepsilon$-coreset $(\mathcal{S}, v)$ with probability at least $1 - \delta$.

// Compute the optimal solution using an Interior Point Method.

1   $w^* \leftarrow \text{InteriorPointMethod}(\mathcal{P}, u, C)$

2   $K_{y_i} \leftarrow u(\mathcal{P}_{y_i}^{\mathsf{c}}) / (2u(\mathcal{P}) \cdot u(\mathcal{P}_{y_i}))$ for each $y_i \in \{-1, 1\}$;

// Compute an upper bound for the sensitivity of each point
     according to Eqn.(5).

3   **for** $i \in [n]$ **do**

4       $p_\Delta \leftarrow (\bar{p}_{y_i} - p_i)$ where $\bar{p}_{y_i} = \frac{1}{u(\mathcal{P}_{y_i})} \sum_{q \in \mathcal{P}_{y_i}} u(q)\, q$

5       $\gamma(p_i) \leftarrow \frac{u(p_i)}{u(\mathcal{P}_{y_i})} + \frac{Cu(p_i)}{2f(\mathcal{P}, w^*)} \left( \sqrt{ \left( \langle w^*, p_\Delta \rangle - \frac{K_{y_i}}{C} \right)^2 + 2f(\mathcal{P}, w^*) \|p_\Delta\|_2^2 } + \langle w^*, p_\Delta \rangle - \frac{K_{y_i}}{C} \right)$

6   $t \leftarrow \sum_{i \in [n]} \gamma(p_i)$

7   Let

$$m \leftarrow \Omega\left( \frac{t}{\varepsilon^2} \big( d \log t + \log(1/\delta) \big) \right)$$

8   $(K_1, \ldots, K_n) \sim \text{Multinomial}\,(m, \pi_i = \gamma(p_i)/t \ \ \forall i \in [n])$

9   $\mathcal{S} \leftarrow \{p_i \in \mathcal{P} \ : \ K_i > 0\}$

// Compute the weights $v : \mathcal{S} \to \mathbb{R}_{\geq 0}$ for every point $p_i \in \mathcal{S}$.

10   **for** $i \in [n]$ **do**

11       $v(p_i) \leftarrow \frac{t K_i u(p_i)}{\gamma(p_i)|\mathcal{S}|}$

12   **return** $(\mathcal{S}, v)$

---

for the existence of small coresets depending on the properties of the data. Our theoretical results also highlight the influence of the regularization parameter, $C$, in the size of the coreset.

**Definition 4** (Sensitivity (Braverman et al., 2016)). *The sensitivity of an arbitrary point $p \in \mathcal{P}$, $p = (x, y)$ is defined as*

$$s(p) = \sup_{w \in \mathcal{Q}} \frac{u(p)f(p, w)}{\sum_{q \in \mathcal{P}} u(q)f(q, w)}, \tag{4}$$

*where $u : \mathcal{P} \to \mathbb{R}_{\geq 0}$ is the weight function as before.*

### 5.1 SENSITIVITY LOWER BOUND

We first prove the existence of a hard point set for which the sum of sensitivities is approximately $\Omega(nC)$, ignoring $d$ factors, which suggests that if the regularization parameter is too large, then the required number of samples for property (3) to hold is $\Omega(n)$.

**Lemma 5.** *There exists a set of $n$ points $\mathcal{P}$ such that the sensitivity of each point $p_i$ is bounded below by $\Omega\left( \frac{d^2 + nC}{n(C + d^2)} \right)$ and the sum of sensitivities is bounded below by $\Omega\left( \frac{d^2 + nC}{C + d^2} \right)$.*

The same hard point set from Lemma 5 can be used to also prove a bound that is nearly exponential in the dimension, $d$.

**Corollary 6.** *There exists a set of $n$ points $\mathcal{P}$ such that the total sensitivity is bounded below by $\Omega\left( \frac{d^2 + C\left( 2^d / \sqrt{d} \right)}{d^2 + C} \right)$.*

We next prove upper bounds on the sensitivity of each data point with respect to the complexity of the input data. Despite the non-existence results established above, our upper bounds shed light into the class of data sets for which coresets of sufficiently small size exist, and thus have potential to significantly speed up SVM training.

## 5.2 Sensitivity Upper Bound

For any arbitrary point $p = (x_i, y_i) \in \mathcal{P}$, let $\mathcal{P}_{y_i} \subset \mathcal{P}$ denote the set of points of label $y_i$, let $\mathcal{P}_{y_i}^{\mathsf{c}} = \mathcal{P} \setminus \mathcal{P}_{y_i}$ be its complement, and let $w^*$ denote the optimal solution to the SVM problem (2). We assume that the points are normalized to have a Euclidean norm of at most one, i.e., $\forall (x, y) \in \mathcal{P} \; \|x_{1:d}\|_2 \leq 1$, where $x_{1:d}$ refers to original input point, without the bias embedding.

**Lemma 7.** *The sensitivity of any point $p_i \in \mathcal{P}$ is bounded above by*

$$s(p_i) \leq \frac{u(p_i)}{u(\mathcal{P}_{y_i})} + \frac{Cu(p_i)}{2f(\mathcal{P}, w^*)} \left( \sqrt{\left( \langle w^*, p_\Delta \rangle - \frac{K_{y_i}}{C} \right)^2 + 2f(\mathcal{P}, w^*) \|p_\Delta\|_2^2} + \langle w^*, p_\Delta \rangle - \frac{K_{y_i}}{C} \right)$$

$$= \gamma(p_i), \tag{5}$$

*where $p_\Delta = \bar{p}_{y_i} - p_i$ and $K_{y_i} = u(\mathcal{P}_{y_i}^{\mathsf{c}}) / (2u(\mathcal{P}) \cdot u(\mathcal{P}_{y_i}))$.*

Let $\mathcal{P}_+ = \mathcal{P}_1 \subset \mathcal{P}$ and $\mathcal{P}_- = \mathcal{P} \setminus \mathcal{P}_1$ denote the set of points with positive and negative labels respectively. Let $\bar{p}_+$ and $\bar{p}_-$ denote the weighted mean of the positive and labeled points respectively, and for any $p_i \in \mathcal{P}_+$ let $p_{\Delta_i}^+ = \bar{p}_+ - p_i$ and $p_{\Delta_i}^- = \bar{p}_- - p_i$.

**Lemma 8.** *The sum of sensitivities over all points $\mathcal{P}$ is bounded by*

$$S(\mathcal{P}) \leq 2 + \frac{C \left( Var(\mathcal{P}_+) + Var(\mathcal{P}_-) \right)}{\sqrt{f(\mathcal{P}, w^*)}} = t, \tag{6}$$

*where $f(\mathcal{P}, w^*)$ is the optimal value of the SVM problem, and $Var(\mathcal{P}_+)$ and $Var(\mathcal{P}_-)$ denote the total deviation of positive and negative labeled points from their corresponding label-specific mean*

$$Var(\mathcal{P}_+) = \sum_{p_i \in \mathcal{P}_+} u(p_i) \left\| p_{\Delta_i}^+ \right\|_2 = \sum_{p_i \in \mathcal{P}_+} u(p_i) \left\| \bar{p}_+ - p_i \right\|_2$$

$$Var(\mathcal{P}_-) = \sum_{p_i \in \mathcal{P}_-} u(p_i) \left\| p_{\Delta_i}^- \right\|_2 = \sum_{p_i \in \mathcal{P}_-} u(p_i) \left\| \bar{p}_- - p_i \right\|_2.$$

**Theorem 9.** *Given any $\varepsilon \in (0, 1/2), \delta \in (0, 1)$ and a weighted data set $\mathcal{P}$ with corresponding weight function $u$, with probability greater than $1 - \delta$, Algorithm 1 generates an $\varepsilon$-coreset, i.e., a weighted set $(\mathcal{S}, v)$, of size*

$$\mathcal{S} \in \Omega \left( \frac{t}{\varepsilon^2} \left( d \log t + \log(1/\delta) \right) \right)$$

*in $\mathcal{O}(n^3)$ time, where $t$ is the upper bound on the sum of sensitivities from Lemma 8,*

$$t = 2 + \frac{C \left( Var(\mathcal{P}_+) + Var(\mathcal{P}_-) \right)}{\sqrt{f(\mathcal{P}, w^*)}}.$$

For any subset $T \subseteq \mathcal{P}$, let $w_T^*$ denote the optimal separating hyperplane with respect to the set of points in $T$. The following corollary immediately follows from Theorem 9 and implies that training an SVM on an $\varepsilon$-coreset, $(\mathcal{S}, v)$, to obtain $w_{\mathcal{S}}^*$ yields a solution that is provably competitive with the optimal solution on the full data-set, $w_{\mathcal{P}}^* = w^*$.

**Corollary 10.** *Given any $\varepsilon \in (0, 1/2), \delta \in (0, 1)$ and a weighted data set $(\mathcal{P}, u)$, the weighted set of points $(\mathcal{S}, v)$ generated by Alg. 1 satisfies*

$$f((\mathcal{P}, u), w_{\mathcal{S}}^*) \leq (1 + 4\varepsilon) f((\mathcal{P}, u), w_{\mathcal{P}}^*),$$

*with probability greater than $1 - \delta$.*

**Sufficient Conditions** Theorem 9 immediately implies that, for reasonable $\varepsilon$ and $\delta$, coresets of polylogarithmic (in $n$) size can be obtained if $d = \mathcal{O}(\text{polylog}(n))$, which is usually the case in our target applications, and if

$$\frac{C \left( \text{Var}(\mathcal{P}_+) + \text{Var}(\mathcal{P}_-) \right)}{\sqrt{f(\mathcal{P}, w^*)}} = \mathcal{O}(\text{polylog}(n)).$$

For example, a value of $C \leq \frac{\log n}{n}$ for the regularization parameter $C$ satisfies the sufficient condition for all data sets with normalized points.

**Interpretation of Bounds**    Our approximation of the sensitivity of a point $p_i \in \mathcal{P}$, i.e., its relative importance, is a function of the following highly intuitive variables.

1. Relative weight with respect to the weights of points of the same label $(u(p_i)/u(\mathcal{P}_{y_i}))$: the sensitivity increases as this ratio increases.
2. Distance to the label-specific mean point $\left(\|\bar{p}_{y_i} - p_i\|_2\right)$: points that are considered outliers with respect to the label-specific cluster are assigned higher importance.
3. Distance to the optimal hyperplane $(\langle w^*, p_\Delta \rangle)$: importance increases as distance of the difference vector $p_\Delta = \bar{p}_{y_i} - p_i$ to the optimal hyperplane increases.

Note that the sum of sensitivities, which dictates how many samples are necessary to obtain an $\varepsilon$-coreset with probability at least $1 - \delta$ and in a sense measures the difficulty of the problem, increases monotonically with the sum of distances of the points from their label-specific means.

### 5.3 EXTENSIONS

We conclude our analysis with an extension of Lemma 7 to the case where only an approximately optimal solution to the SVM problem is available.

**Lemma 11.** *Consider the case where only a $\xi$-approximate solution $\hat{w}$ is available such that $f(\mathcal{P}, \hat{w}) \leq f(\mathcal{P}, w^*) + \xi$, for $\xi \in (0, f(\mathcal{P}, w^*)/2)$. Then, the sensitivity of any arbitrary point $p_i \in \mathcal{P}$ is bounded above by*

$$s(p_i) \leq \frac{u(p_i)}{u(\mathcal{P}_{y_i})} + \frac{Cu(p_i)\left(\sqrt{\left(\langle \hat{w}, p_\Delta \rangle - \frac{K_{y_i}}{C}\right)^2 + 4\left(f(\mathcal{P}, \hat{w}) - 2\xi\right)\|p_\Delta\|_2^2} + \langle \hat{w}, p_\Delta \rangle - \frac{K_{y_i}}{C}\right)}{2\left(f(\mathcal{P}, \hat{w}) - 2\xi\right)},$$

*where $p_\Delta = \bar{p}_{y_i} - p_i$ and $K_{y_i} = u(\mathcal{P}_{y_i}^{\mathsf{c}})/\left(2u(\mathcal{P}) \cdot u(\mathcal{P}_{y_i})\right)$ as in Lemma 7.*

## 6 RESULTS

We evaluate the performance of our coreset construction algorithm against synthetic and real-world, publicly available data sets (Lichman, 2013). We compare the effectiveness of our method to uniform subsampling on a wide variety of data sets and also to Pegasos, one of the most popular Stochastic-Gradient Descent based algorithm for SVM training (Shalev-Shwartz et al., 2011). For each data set of size $N$, we selected a set of $M = 10$ subsample sizes $S_1, \ldots, S_M \subset [N]$ and ran each coreset construction algorithm to construct and evaluate the accuracy subsamples sizes $S_1, \ldots, S_M$. The results were averaged across 100 trials for each subsample size. Our results of relative error and sampling variance are shown as Figures 1 and 3. The computation time required for each sample size and approach can be found in the Appendix (Fig. 4). Our experiments were implemented in Python and performed on a 3.2GHz i7-6900K (8 cores total) machine with 16GB RAM. We considered the following data sets for evaluation.

- *Pathological* — $1,000$ points in two dimensional space describing two clusters distant from each other of different labels, as well as two points of different labels which are close to each other. We note that uniform sampling performs particularly poorly against this data set due to the presence of outliers.
- *Synthetic & Synthetic100K* — The Synthetic and Synthetic100K are datasets with $6,000, 100,000$ points, each consisting of 3 and 4 dimensions respectively. The datasets describe two blocks of mirrored nested rings of points, each of different labels such that Gaussian noise has been added to them.
- *HTRU²* — $17,898$ radio emissions of Pulsar (rare type of Neutron star) each consisting of 9 features.
- *CreditCard* [3] — $30,000$ client entries each consisting of 24 features that include education, age, and gender among other factors.

---

[2]`https://archive.ics.uci.edu/ml/datasets/HTRU2`
[3]`https://archive.ics.uci.edu/ml/datasets/default+of+credit+card+clients`

- *Skin*[4] — $245,057$ random samples of B,G,R from face images consisting of 4 dimensions.

**Evaluation** We computed the relative error of the sampling-based algorithms with respect to the cost of the optimal solution to the SVM problem, $f(\mathcal{P}, w_{\mathcal{P}}^*)$ and the approximate cost generated by the subsample, $f((\mathcal{S}, v), w_{\mathcal{S}}^*)$. We have also evaluated against Pegasos, running Pegasos the amount of time needed to construct the coreset and comparing the resulted error, applying 128 repetitions as presented at Figure 2. Furthermore, we have run our coreset constructing under streaming setting, where subsamples are used as leaf size and half of the leaf's size is then used to set the subsample for our sampling approach. In addition, we also compared our coreset construction's related error to CVM's related error with respect to the cost of the optimal solution to the SVM problem, as function of subsample sizes. Finally, we have evaluated the variance of the estimators for the sampling-based approaches and observed that the variances of the estimates generated by our coreset were lower than those of uniform subsampling.

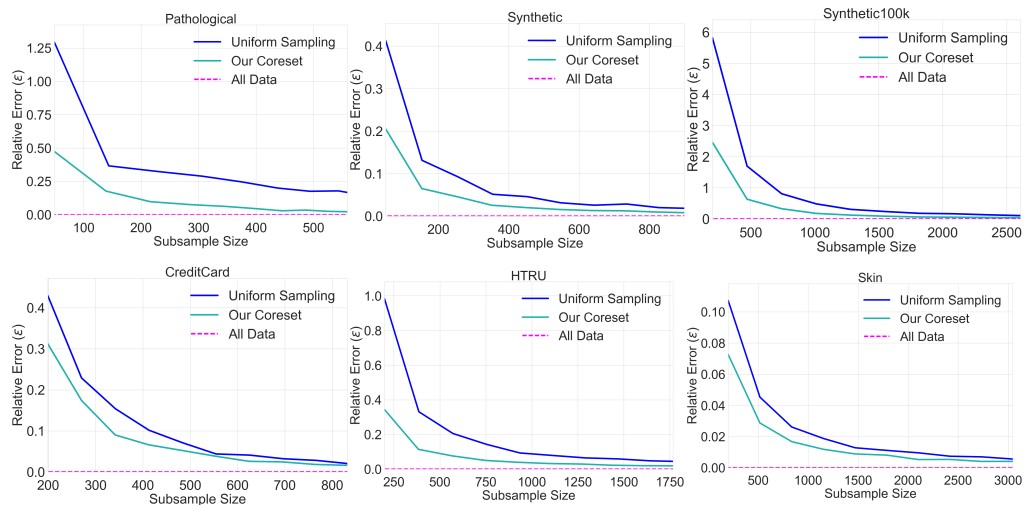

Figure 1: The relative error of query evaluations with respect uniform and coreset subsamples for the 4 data sets.

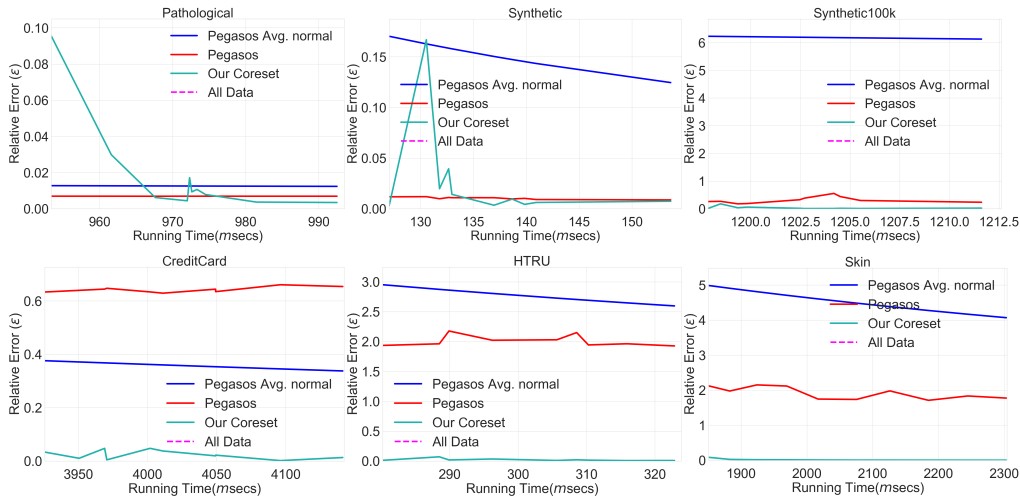

Figure 2: The relative error of query evaluations with respect to coreset running time and Pegasos running time for the 4 data sets.

---

[4] https://archive.ics.uci.edu/ml/datasets/Skin+Segmentation/

# 7 CONCLUSION

We presented an efficient coreset construction algorithm for generating compact representations of the input data points that provide provably accurate inference. We presented both lower and upper bounds on the number of samples required to obtain accurate approximations to the SVM problem as a function of input data complexity and established sufficient conditions for the existence of compact representations. Our experimental results demonstrate the effectiveness of our approach in speeding up SVM training when compared to uniform sub-sampling

The method presented in this paper is also applicable to streaming settings, using the merge-and-reduce technique from coresets literature (Braverman et al., 2016).We conjecture that our coreset construction method can be extended to significantly speed up SVM training for nonlinear kernels as well as other popular machine learning algorithms, such as deep learning.

# 8 APPENDIX

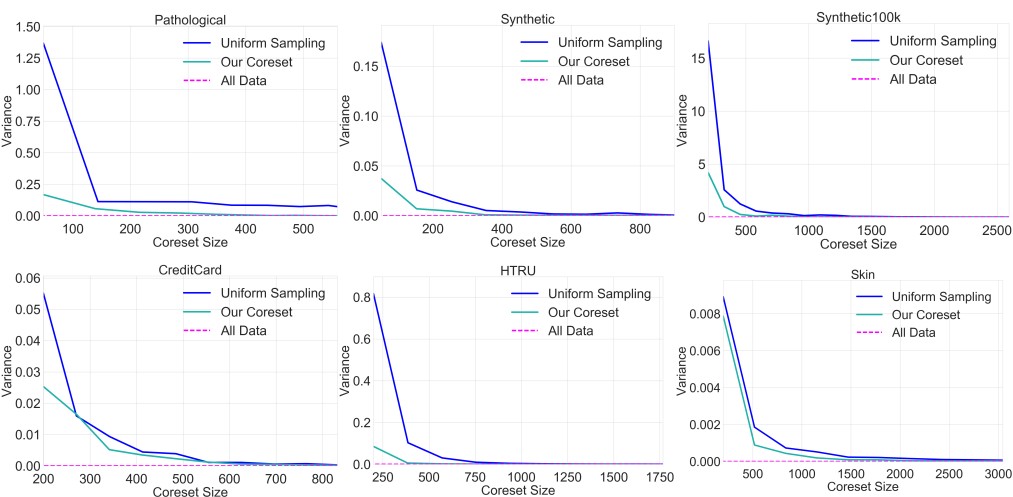

Figure 3: The estimator variance of query evaluations. We note that due to the use of a judicious sampling distribution based on the points' sensitivities, the variance of our coreset estimator is lower than that of uniform sampling for all data sets.

## 8.1 PROOF OF LEMMA 5

*Proof.* Following Yang et al. (2017) we define the set of $n$ points $\mathcal{P} \subseteq \mathbb{R}^{d+1} \times \{-1, 1\}$, each point $p \in \mathcal{P}$ with weight $u(p) = 1$, such that for each $i \in [n]$, among the first $d$ entries of $p_i$, exactly $d/2$ entries are equivalent to $\gamma$:

$$\gamma = \sqrt{\frac{R+1}{d}},$$

where $R$ is the normalization factor (typically $R = 1$), the remaining $d/2$ entries among the first $d$ are set to 0, and $p_{i(d+1)} = y_i$ as before. For each $i \in [n]$, define the set of non-zero entries of $p_i$ as the set

$$B_i = \{j \in [d+1] : p_{ij} \neq 0\}.$$

Now, for bounding the sensitivity of point $p_i$, consider the normal to the margin $w_i$ with entries defined as

$$\forall j \in [d+1] \quad w_{ij} = \begin{cases} 0 & \text{if } \ j \in B_i, \\ 1/\gamma & \text{otherwise.} \end{cases}$$

Note that for $R = 1$, $||w_i||^2 = (d/2)(1/\gamma^2) = (d/2)d/(R+1) = d^2/4$. We also have that $h(w_i, p_i) = 1$ since $p_i \cdot w_i = \sum_{l \in B_i} p_{ij}w_{ij} = (d/2)(0) = 0$. To bound the sum of hinge losses

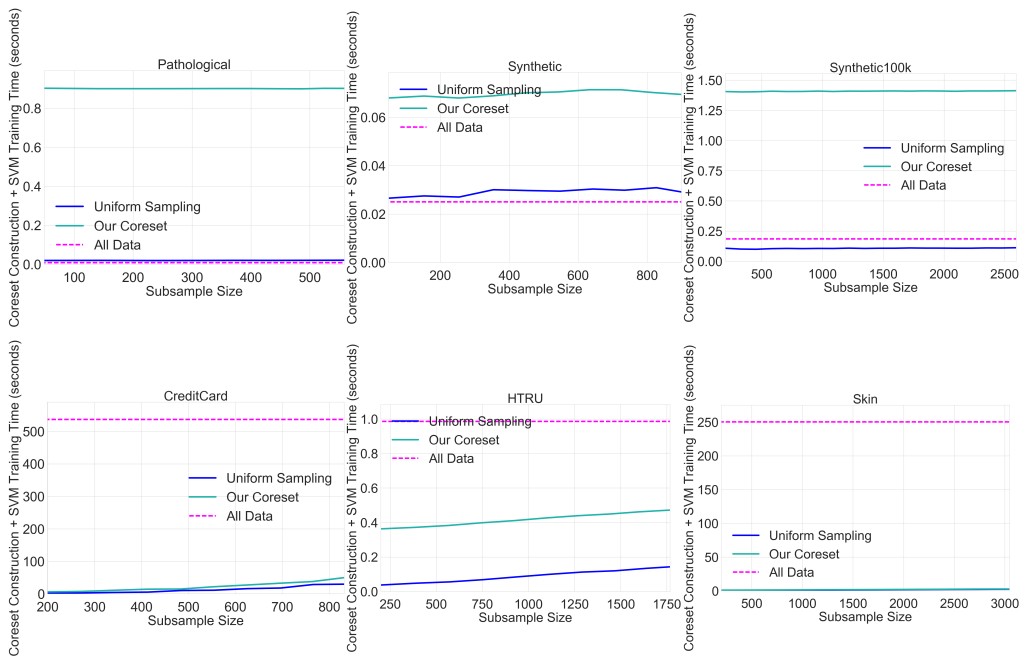

Figure 4: Computation + Training Time vs. Sample Size

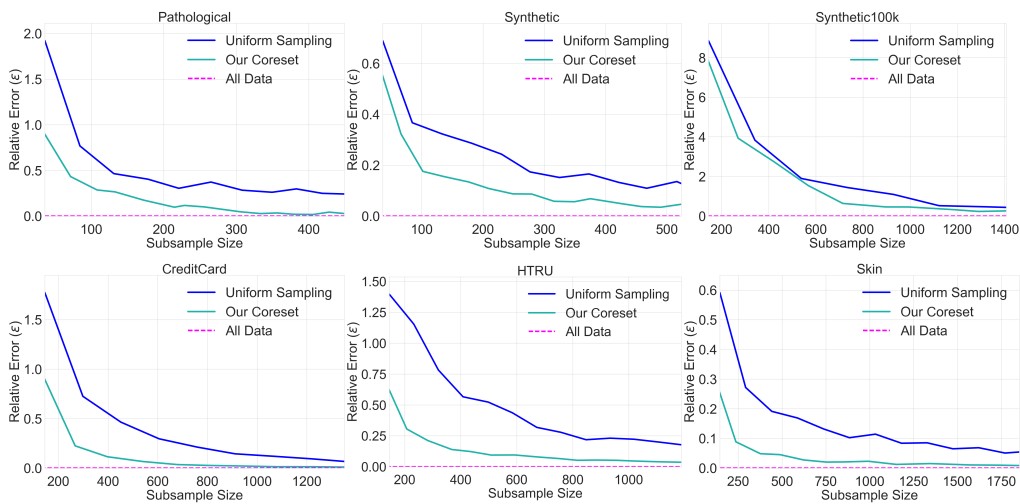

Figure 5: The relative error of query evaluations with respect uniform and coreset subsamples for the 5 data sets in a streaming setting where the input points arrive one-by-one.

contributed by other points $j \in [n]$, $j \neq i$ note that $B_i \setminus B_j \neq \emptyset$, thus:

$$\langle w_i, p_j \rangle = \sum_{l \in B_i \setminus B_j} w_{il} p_{jl} \geq \gamma(1/\gamma) = 1,$$

which implies that $h(p_j, w_i) = 0$. Thus, it follows that $H(w_i) = \sum_{j \in [n]} h(p_j, w_i) = 1$.

Putting it all together, we have for the sensitivity of any arbitrary $i \in [n]$:

$$s(p_i) = \sup_{w \in \mathcal{Q}} \frac{f(p_i, w)}{\sum_{j \in [n]} f(p_j, w)} \geq \frac{\frac{d^2}{8n} + C\, h(p_i, w_i)}{\frac{||w_i||^2}{2} + C} = \frac{\frac{d^2}{8n} + C}{\frac{d^2}{8} + C}.$$

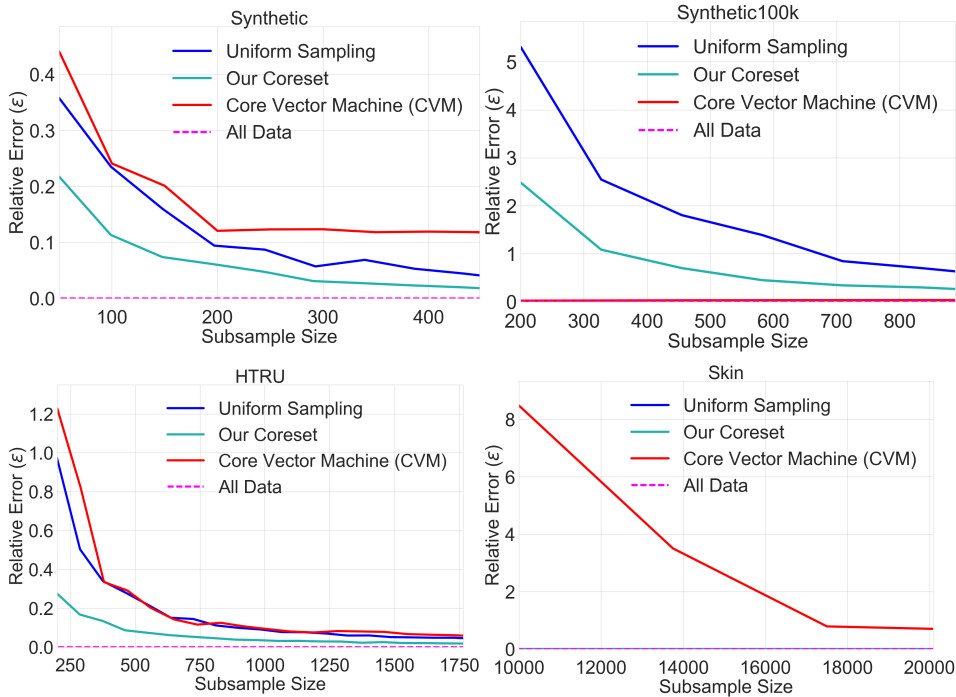

Figure 6: The relative error of query evaluations with respect uniform, coreset, and Core Vector Machine (CVM) (Tsang et al., 2005) subsamples for the 4 data sets.

Moreover, we have for the sum of sensitivities that

$$
\sum_{i \in [n]} s(p_i) \geq \frac{\frac{d^2}{8} + nC}{\frac{d^2}{8} + C} = \Omega\left(\frac{d^2 + nC}{d^2 + C}\right)
$$

□

## 8.2 PROOF OF COROLLARY 6

*Proof.* Consider the set of points $\mathcal{P}$ from the proof of Lemma 5 and note that

$$
n = \binom{d}{d/2} \geq \frac{2^d(1 - 1/d)}{\sqrt{\pi(d/2)}} \geq \frac{2^d}{\sqrt{d}} = \Omega(2^d/\sqrt{d}).
$$

□

## 8.3 PROOF OF LEMMA 7

*Proof.* Consider any arbitrary point $p_i \in \mathcal{P}$ and let $p = x_i y_i$ for brevity when the point $p_i = (x_i, y_i)$ is clear from the context. We proceed to bound $s(p_i)/u(p_i)$ by first leveraging the Lipschitz

continuity of hinge loss and convexity of $f$

$$
\begin{aligned}
\frac{s(p_i)}{u(p_i)} &= \sup_{w \in Q(\mathcal{P})} \frac{f(p_i, w)}{f(\mathcal{P}, w)} \\
&\leq \sup_{w \in Q} \frac{f(\bar{p}_{y_i}, w) + C\left[\langle w, \bar{p}_{y_i} - p\rangle\right]_+}{f(\mathcal{P}, w)} && \text{Lipschitz Continuity} \\
&\leq \sup_{w \in Q} \frac{\sum_{q \in \mathcal{P}_{y_i}} u(q) f(q, w)}{u(\mathcal{P}_{y_i}) f(\mathcal{P}, w)} + \frac{C\left[\langle w, \bar{p}_{y_i} - p\rangle\right]_+}{f(\mathcal{P}, w)} && \text{Jensen's Inequality} \\
&= \sup_{w \in Q} \frac{\left(f(\mathcal{P}_{y_i}, w) + f(\mathcal{P}_{y_i}^{\mathsf{c}}, w)\right) - f(\mathcal{P}_{y_i}^{\mathsf{c}}, w)}{u(\mathcal{P}_{y_i}) f(\mathcal{P}, w)} + \frac{C\left[\langle w, \bar{p}_{y_i} - p\rangle\right]_+}{f(\mathcal{P}, w)} \\
&= \frac{1}{u(\mathcal{P}_{y_i})} + \sup_{w \in Q} \frac{C\left[\langle w, \bar{p}_{y_i} - p\rangle\right]_+}{f(\mathcal{P}, w)} - \frac{f(\mathcal{P}_{y_i}^{\mathsf{c}}, w)}{u(\mathcal{P}_{y_i}) f(\mathcal{P}, w)} \\
&\leq \frac{1}{u(\mathcal{P}_{y_i})} + \sup_{w \in Q} \frac{C\left[\langle w, \bar{p}_{y_i} - p\rangle\right]_+ - u(\mathcal{P}_{y_i}^{\mathsf{c}})/(2u(\mathcal{P}) \cdot u(\mathcal{P}_{y_i}))}{f(\mathcal{P}, w)}, && (7)
\end{aligned}
$$

where the last inequality follows from the fact that for any $w \in Q$, $\|w_{1:d}\|_2 \geq 1$ since the points are normalized, and thus,

$$
\frac{f(\mathcal{P}_{y_i}^{\mathsf{c}}, w)}{2u(\mathcal{P}_{y_i})} \geq \frac{\|w_{1:d}\|_2^2 \, u(\mathcal{P}_{y_i}^{\mathsf{c}})}{2u(\mathcal{P}) u(\mathcal{P}_{y_i})} \geq u(\mathcal{P}_{y_i}^{\mathsf{c}})/\left(2u(\mathcal{P}) \cdot u(\mathcal{P}_{y_i})\right).
$$

Now consider the expression involving the supremum from (7) and let

$$
K_{y_i} = u(\mathcal{P}_{y_i}^{\mathsf{c}})/\left(2u(\mathcal{P}) \cdot u(\mathcal{P}_{y_i})\right) \tag{8}
$$

be the constant independent of $w$ in the numerator, let $p_\Delta = \bar{p}_{y_i} - p$, and let $w_\Delta = w - w^*$ for notational convenience. We observe that by $\lambda$-strong convexity of the SVM objective function for $\lambda = 1$, we have the inequality $f(\mathcal{P}, w^*) + \lambda \|w - w^*\|_2^2 / 2 \leq f(\mathcal{P}, w)$ for any $w \in Q$. Thus, the expression containing the supremum from (7) is bounded by

$$
\begin{aligned}
\sup_{w \in Q} \frac{C\left[\langle w, \bar{p}_{y_i} - p\rangle\right]_+ - K_{y_i}}{f(\mathcal{P}, w)} &= \sup_{w \in Q} \frac{C\left[\langle w - w^*, \bar{p}_{y_i} - p\rangle + \langle w^*, \bar{p}_{y_i} - p\rangle\right]_+ - K_{y_i}}{f(\mathcal{P}, w)} \\
&\leq \sup_{w_\Delta \in Q} \frac{C\left[\langle w_\Delta, p_\Delta\rangle + \langle w^*, p_\Delta\rangle\right]_+ - K_{y_i}}{f(\mathcal{P}, w^*) + \lambda \|w_\Delta\|_2^2 / 2} \\
&\leq \sup_{w_\Delta \in Q} \frac{\left[\langle w_\Delta, p_\Delta\rangle + a\right]_+ + b}{c \|w_\Delta\|_2^2 + d}, && (9)
\end{aligned}
$$

where $a = \langle w^*, p_\Delta\rangle$, $b = -K_{y_i}/C$, $c = 1/(2C)$, and $d = f(P, w^*)/C$ are constants independent of $w_\Delta$. Analytical optimization based on the gradient of the term above yields the optimal solution, $w_\Delta^* = \frac{p_\Delta}{2cz}$, where

$$
z = \frac{\langle w_\Delta^*, p_\Delta\rangle + a + b}{c \|w_\Delta^*\|_2^2 + d}. \tag{10}
$$

We resolve the interdependency on $w_\Delta^*$ by observing that

$$
\langle w_\Delta^*, p_\Delta\rangle = \left\langle \frac{p_\Delta}{2cz}, p_\Delta \right\rangle = \frac{\|p_\Delta\|_2^2}{2cz} \quad \text{and} \quad \|w_\Delta^*\|_2^2 = \langle w_\Delta^*, w_\Delta^*\rangle = \frac{\|p_\Delta\|_2^2}{(2cz)^2},
$$

which implies that $\|w_\Delta^*\|_2^2 = \langle w_\Delta^*, p_\Delta\rangle/(2cz)$. Putting it all together, we obtain

$$
\begin{aligned}
z &\leq \frac{\sqrt{(a+b)^2 + \frac{d\|p_\Delta\|_2^2}{c}} + a + b}{2d} \\
&= \frac{C}{2f(\mathcal{P}, w^*)} \left( \sqrt{(\langle w^*, p_\Delta\rangle - K_{y_i}/C)^2 + 2f(\mathcal{P}, w^*) \|p_\Delta\|_2^2} + \langle w^*, p_\Delta\rangle - K_{y_i}/C \right),
\end{aligned}
$$

and the sensitivity bound follows. $\qquad\square$

## 8.4 Proof of Lemma 8

*Proof.* Let $\mathcal{P}_+ = \mathcal{P}_1 \subset \mathcal{P}$ and $\mathcal{P}_- = \mathcal{P} \backslash \mathcal{P}_1$ denote the set of points with positive and negative labels respectively and let $K_+$ and $K_-$ denote the corresponding constants defined by (8). Let $\bar{p}_+$ and $\bar{p}_-$ denote the weighted mean of the positive and labeled points respectively, and for any $p_i \in \mathcal{P}_+$ let $p_{\Delta_i}^+ = \bar{p}_+ - p_i$ and $p_{\Delta_i}^- = \bar{p}_- - p_i$.

Since the sensitivity can be decomposed into sum over the two disjoint sets, i.e., $S(\mathcal{P}) = \sum_{p \in \mathcal{P}} s(p) = \sum_{p \in \mathcal{P}_1} s(p) + \sum_{p \in \mathcal{P}_-} s(p) = S(\mathcal{P}_1) + S(\mathcal{P}_-)$, we consider first bounding $S(\mathcal{P}_1)$. Invoking Lemma 7 yields

$$S(\mathcal{P}_1) \leq 1 + \sum_{p_i \in \mathcal{P}_+} \frac{Cu(p_i)}{2f(\mathcal{P}, w^*)} \left( \sqrt{\left( \langle w^*, p_{\Delta_i}^+ \rangle - K_+/C \right)^2 + 2f(\mathcal{P}, w^*) \left\| p_{\Delta_i}^+ \right\|_2^2} + \langle w^*, p_{\Delta_i}^+ \rangle - K_+/C \right)$$

$$\leq 1 + \sum_{p_i \in \mathcal{P}_+} \frac{Cu(p_i)}{2f(\mathcal{P}, w^*)} \left( \sqrt{\left( \langle w^*, p_{\Delta_i}^+ \rangle - K_+/C \right)^2 + 2f(\mathcal{P}, w^*) \left\| p_{\Delta_i}^+ \right\|_2^2} \right)$$

$$\leq 1 + \frac{C}{f(\mathcal{P}, w^*)} \sum_{p_i \in \mathcal{P}_+} u(p_i) \left( \left\| p_{\Delta_i}^+ \right\|_2 \sqrt{f(\mathcal{P}, w^*)} \right)$$

$$= 1 + \frac{C}{\sqrt{f(\mathcal{P}, w^*)}} \sum_{p_i \in \mathcal{P}_+} u(p_i) \left\| p_{\Delta_i}^+ \right\|_2,$$

where the second equality follows by the definition of $K_+$ and the fact that

$$\sum_{p_i \in \mathcal{P}_+} u(p_i) \langle w^*, p_{\Delta_i}^+ \rangle = \sum_{p_i \in \mathcal{P}_+} u(p_i) \langle w^*, \bar{p}_+ \rangle - \sum_{p_i \in \mathcal{P}_+} u(p_i) \langle w^*, p_i \rangle$$

$$= \langle w^*, \bar{p}_+ \rangle \sum_{p_i \in \mathcal{P}_+} u(p_i) - \sum_{p_i \in \mathcal{P}_+} u(p_i) \langle w^*, p_i \rangle = 0,$$

and the third by noting that $(\langle w^*, p_{\Delta_i}^+ \rangle - K_+/C)^2 \leq 2f(\mathcal{P}, w^*) \left\| p_{\Delta_i}^+ \right\|_2^2$ since by Cauchy-Schwarz and definition of $f(\mathcal{P}, w^*)$ we have

$$\langle w^*, p_{\Delta_i}^+ \rangle - K_+/C \leq \|w^*\|_2 \left\| p_{\Delta_i}^+ \right\|_2 \leq \sqrt{2f(\mathcal{P}, w^*)} \left\| p_{\Delta_i}^+ \right\|_2.$$

Using the same argument as above, an analogous bound holds for $S(\mathcal{P}_-)$, thus we have

$$S(\mathcal{P}) \leq 2 + \frac{C}{\sqrt{f(\mathcal{P}, w^*)}} \left( \sum_{p_i \in \mathcal{P}_1} u(p_i) \left\| p_{\Delta_i}^+ \right\|_2 + \sum_{p_i \in \mathcal{P}_-} u(p_i) \left\| p_{\Delta_i}^- \right\|_2 \right)$$

$$= 2 + \frac{C \left( \text{Var}(\mathcal{P}_+) + \text{Var}(\mathcal{P}_-) \right)}{\sqrt{f(\mathcal{P}, w^*)}} = t.$$

$\square$

## 8.5 Proof of Theorem 9

*Proof.* By Lemma 8 and Theorem 5.5 of Braverman et al. (2016) we have that the coreset constructed by our algorithm is an $\varepsilon$-coreset with probability at least $1 - \delta$ if

$$|\mathcal{S}| \geq \Omega \left( \frac{t}{\varepsilon^2} \left( d \log t + \log(1/\delta) \right) \right),$$

where we used the fact that the VC dimension of a separating hyperplane in the case of a linear kernel is bounded $\dim(\mathcal{F}) \leq d + 1 = \mathcal{O}(d)$ (Vapnik & Vapnik, 1998). Moreover, note that the computation time of our algorithm is dominated by computing the optimal solution of the SVM problem using interior-point Method which takes $\mathcal{O}(d^3 L) = \mathcal{O}(n^3)$ time (Nesterov & Nemirovskii, 1994), where $L$ is the bit length of the input data. $\square$

## 8.6 PROOF OF COROLLARY 10

*Proof.* By theorem 9, $(\mathcal{S}, v)$ is an $\varepsilon$-coreset for $(\mathcal{P}, u)$ with probability at least $1 - \delta$, thus we have

$$f((\mathcal{P}, u), w_{\mathcal{P}}^*) \leq f((\mathcal{P}, u), w_{\mathcal{S}}^*) \leq \frac{f((\mathcal{S}, v), w_{\mathcal{S}}^*)}{1 - \varepsilon} \leq \frac{(1 + \varepsilon)f((\mathcal{P}, u), w_{\mathcal{P}}^*)}{1 - \varepsilon}$$

$$\leq (1 + 4\varepsilon)f((\mathcal{P}, u), w_{\mathcal{P}}^*).$$

$\square$

## 8.7 PROOF OF LEMMA 11

*Proof.* The proof is almost identical to that of Lemma 7, thus we use the same techniques to arrive at the following inequality with $\hat{w}$ instead of $w^*$ and with $w_\Delta = w - \hat{w}$:

$$\frac{s(p_i)}{u(p_i)} = \sup_{w \in Q(\mathcal{P})} \frac{f(p_i, w)}{f(\mathcal{P}, w)}$$

$$\leq \frac{1}{u(\mathcal{P}_{y_i})} + \sup_{w \in \mathcal{Q}} \frac{C\left[\langle w, \bar{p}_{y_i} - p\rangle\right]_+ - u(\mathcal{P}_{y_i}^{\mathsf{c}})/(2u(\mathcal{P}) \cdot u(\mathcal{P}_{y_i}))}{f(\mathcal{P}, w)}$$

$$\leq \frac{1}{u(\mathcal{P}_{y_i})} + \sup_{w_\Delta \in \mathcal{Q}} \frac{C\left[\langle w_\Delta, p_\Delta\rangle + \langle \hat{w}, p_\Delta\rangle\right]_+ - K_{y_i}}{f(\mathcal{P}, w^*) + \|w - w^*\|_2^2/2}.$$

Now, consider the supremum term and note that (i) $f(\mathcal{P}, w^*) \geq f(\mathcal{P}, \hat{w}) - \xi$ by definition of $\hat{w}$ and (ii) $\|w - w^*\|_2^2/2 \geq \|w - \hat{w}\|_2^2/4 - \xi$, since by the Cauchy-Schwarz inequality we have

$$\|w - \hat{w}\|_2^2 \leq 2\left(\|w - w^*\|_2^2 + \|w^* - \hat{w}\|_2^2\right)$$

$$\leq 2\|w - w^*\|_2^2 + 4\xi,$$

where the last inequality follows by strong convexity of $f$ and by definition of our approximation:

$$\frac{\|\hat{w} - w^*\|_2^2}{2} \leq f(\mathcal{P}, \hat{w}) - f(\mathcal{P}, w^*) \leq \xi.$$

Combining (i) and (ii) yields for the supremum term

$$\sup_{w_\Delta \in \mathcal{Q}} \frac{C\left[\langle w_\Delta, p_\Delta\rangle + \langle \hat{w}, p_\Delta\rangle\right]_+ - K_{y_i}}{f(\mathcal{P}, w^*) + \|w - w^*\|_2^2/2}$$

$$\leq \sup_{w_\Delta \in \mathcal{Q}} \frac{C\left[\langle w_\Delta, p_\Delta\rangle + \langle \hat{w}, p_\Delta\rangle\right]_+ - K_{y_i}}{f(\mathcal{P}, \hat{w}) - 2\xi + \|w_\Delta\|_2^2/4}$$

$$\leq \sup_{w_\Delta \in \mathcal{Q}} \frac{\left[\langle w_\Delta, p_\Delta\rangle + a\right]_+ + b}{c\|w_\Delta\|_2^2 + d},$$

where $a = \langle \hat{w}, p_\Delta\rangle$, $b = -K_{y_i}/C$, $c = 1/(4C)$, and $d = (f(\mathcal{P}, \hat{w}) - 2\xi)/C$ are constants independent of $w_\Delta$.

This is the exact same expression as in the proof of Lemma 7, with the exception of slightly different values for the constants above. Thus, analytical optimization yields the same optimal solution as before and thus we have

$$\frac{s(p_i)}{u(p_i)} \leq \frac{1}{u(\mathcal{P}_{y_i})} + \frac{\sqrt{(a + b)^2 + \frac{d\|p_\Delta\|_2^2}{c}} + a + b}{2d},$$

and the lemma follows. $\square$

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
