# OpenReview forum: "Small Coresets to Represent Large Training Data for Support Vector Machines"
_ICLR.cc/2018/Conference — Reject_

### Official Review · AnonReviewer3 · 2017-11-26

**Rating:** 5
**Confidence:** 3

**Review:**

This paper studies the approach of coreset for SVM. In particular, it aims at sampling a small set of weighted points such that the loss function over these points provably approximates that over the whole dataset. This is done by applying an existing theoretical framework to the SVM training objective.

The coreset idea has been applied to SVM in existing work, but this paper uses a new theoretical framework. It also provides lower bound on the sample complexity of the framework for general instances and provides upper bound that is data dependent, shedding light on what kind of data this method is suitable for.

The main concern I have is about the novelty of the coreset idea applied to SVM. Also, there are some minor issues:
-- Section 4.2: What's the point of building the coreset if you've got the optimal solution? Indeed one can do divide-and-conquer. But can one begin with an approximation solution? In general, the analysis of the coreset should still hold if one begins with an approximation solution. Also, even when doing divide-and-conquer, the solution obtained in the first line of the algorithm should still be approximate. The authors pointed out that Lemma 7 can be extended to this case, and I hope the proof can be written out explicitly.
-- section 2, paragraph 4: why SGD-based approaches cannot be trivially extended to streaming settings?
-- Definition 3: what randomness is the probability with respect to?
-- For experiments: the comparison with CVM should be added.

---

> ### Author Response · Authors · 2017-12-11
> **Response to AnonReviewer3**
>
> Thank you for your comments and feedback. Please find below our item-specific responses.
>
> 1) As we also highlighted in our general response and response to AnonReviewer1, our coreset construction method is intended to be used in conjunction with the traditional merge-and-reduce technique, which ensures that our coreset construction algorithm is never run against the full data set. Rather, our coreset construction algorithm takes as input partitions of the original data set (where each set in the partition is of sufficiently small size, see Sec. 4.2). We have also included an extension of Lemma 7 to the case where only an approximately-optimal solution is available (see Lemma 11 in our revision).
>
> 2) Gradient-based methods cannot be trivially extended to settings where the data points arrive in a streaming fashion, since seeing a new point results in a complete change of the gradient.
>
> 3) Thank you for pointing out this ambiguity. The randomness is with respect to the sampling scheme used by our algorithm to construct the coreset, but we realize in retrospect that this is confusing since there exists deterministic coreset-construction algorithms. We have modified our paper to clarify the definition of a coreset and the (probabilistic) guarantees provided by our algorithm.
>
> 4) As we mentioned to AnonReviewer2, we are currently in the process of running additional experiments that evaluate the performance of our algorithm against other algorithms, such as CVM as you mentioned. Our plan is to include the results of these experiments in a later revision to be uploaded before Dec. 20.

---

### Official Review · AnonReviewer2 · 2017-11-27
**Small Coresets to Represent Large Training Data for Support Vector Machines**

**Rating:** 7
**Confidence:** 3

**Review:**

The paper suggests an importance sampling based Coreset construction for Support Vector Machines (SVM). To understand the results, we need to understand Coreset and importance sampling:

Coreset: In the context of SVMs, a Coreset is a (weighted) subset of given dataset such that for any linear separator, the cost of the separator with respect to the given dataset X is approximately (there is an error parameter \eps) the same as the cost with respect to the weighted subset. The main idea is that if one can find a small coreset, then finding the optimal separator (maximum margin etc.) over the coreset might be sufficient. Since the computation is done over a small subset of points, one hopes to gain in terms of the running time.

Importance sampling: This is based on the theory developed in Feldman and Langberg, 2011 (and some of the previous works such as Langberg and Schulman 2010, the reference of which is missing). The idea is to define a quantity called sensitivity of a data-point that captures how important this datapoint is with respect to contributing to the cost function. Then a subset of datapoint are sampled based on the sensitivity and the sampled data point is given weight proportional to inverse of the sampling probability. As per the theory developed in these past works, sampling a subset of size proportional to the sum of sensitivities gives a coreset for the given problem.

So, the main contribution of the paper is to do all the sensitivity calculations with respect to SVM problem and then use the importance sampling theory to obtain bounds on the coreset size. One interesting point of this construction is that Coreset construction involves solving the SVM problem on the given dataset which may seem like beating the purpose. However, the authors note that one only needs to compute the Coreset of small batches of the given dataset and then use standard procedures (available in streaming literature) to combine the Coresets into a single Coreset. This should give significant running time benefits. The paper also compares the results against the simple procedure where a small uniform sample from the dataset is used for computation.


Evaluation:
Significance: Coresets give significant running time benefits when working with very big datasets. Coreset construction in the context of SVMs is a relevant problem and should be considered significant.

Clarity: The paper is reasonably well-written. The problem has been well motivated and all the relevant issues point out for the reader. The theoretical results are clearly stated as lemmas a theorems that one can follow without looking at proofs.

Originality: The paper uses previously developed theory of importance sampling. However, the sensitivity calculations in the SVM context is new as per my knowledge. It is nice to know the bounds given in the paper and to understand the theoretical conditions under which we can obtain running time benefits using corsets.

Quality: The paper gives nice theoretical bounds in the context of SVMs. One aspect in which the paper is lacking is the empirical analysis. The paper compares the Coreset construction with simple uniform sampling. Since Coreset construction is being sold as a fast alternative to previous methods for training SVMs, it would have been nice to see the running time and cost comparison with other training methods that have been discussed in section 2.

---

> ### Author Response · Authors · 2017-12-11
> **Response to AnonReviewer2**
>
> Thank you for your in-depth feedback and consideration of our paper. We included the reference to the original Langberg and Schulman (2010) paper that introduced the concept of sensitivity. We are currently in the process of running additional experiments that evaluate the performance of our algorithm against larger data-sets and compare it to more of the other approaches mentioned in Sec. 2. We plan to finalize these experiments and include the results in our revised version before Dec. 20.

---

### Official Review · AnonReviewer1 · 2017-12-01
**coreset for svm.**

**Rating:** 5
**Confidence:** 4

**Review:**

The paper studies the problem of constructing small coreset for SVM.
A coreset is a small subset of (weighted) points such that the optimal solution for the coreset is also a good approximation for the original point set. The notion of coreset was originally formulated in computational geometry by Agarwal et al.
(see e.g., [A])
Recently it has been extended to several clustering problems, linear algebra, and machine learning problems. This paper follows the important sampling approach first proposed in [B], and generalized by Feldman and Langberg. The key in this approach is to compute the sensitivity of points and bound the total sensitivity for the considered problem (this is also true for the present paper). For SVM, the paper presents a bad instance where the total sensitivity can be as bad as 2^d. Nevertheless,
the paper presents interesting upper bounds that depending on the optimal value and variance of the point set. The paper argues that in many data sets, the total sensitivity may be small, yielding small coreset. This makes sense and may have significant practical implications.

However, I have the following reservation for the paper.
(1) I don't quite understand the CHICKEN and EGG section. Indeed, it is unclear to me
how to estimate the optimal value. The whole paragraph is hand-waving. What is exactly merge-and-reduce? From the proof of theorem 9, it appears that the interior point algorithm is run on the entire dataset, with running time O(n^3d). Then there is no point to compute a coreset as the optimal solution is already known.

(2) The running time of the algorithm is not attractive (in both theory and practice).
In fact, the experimental result on the running time is a bit weak. It seems that the algorithm is pretty slow (last in Figure 1).

(3) The theoretical novelty is limited. The paper follows from now-standard technique for constructing coreset.

Overal, I don't recommend acceptance.

minor points:
It makes sense to cite the following papers where original ideas on constructing coresets were proposed initially.

[A]Geometric Approximation via Coresets
Pankaj K. Agarwal Sariel Har-Peled Kasturi R. Varadarajan

[B]Universal epsilon-approximators for integrals, by Langberg and Schulman

---------------------------------------------------------

After reading the response and the revised text, I understand the chicken-and-egg issue.
I think the experimental section is still a bit weak (given that there are several very competitive SVM algorithms that the paper didn't compare with).
I raised my score to 5.

---

> ### Author Response · Authors · 2017-12-11
> **Response to AnonReviewer1**
>
> Thank you for your insightful feedback and references to prior work that originally proposed the notion of constructing coresets. We added these references to the revised version of our paper. More specific responses below:
>
> 1) The chicken and the egg phenomena commonly arises in coresets-related work, where the optimal or approximately-optimal solution is used to compute approximations of the sensitivity of each point. In our case, we compute the optimal solution to the problem using the Interior Point Method, but as mentioned in Sec. 4.3, an approximately optimal solution can be computed using Pegasos (Shalev-Shwatz et al., 2011). The merge-and-reduce procedure (explained below and cited in our original submission) ensures that our algorithm is never run against the entire data set, but rather small partitions of the data set. This implies that by repeatedly running our algorithm on a partition of the data set, consisting of approximately logarithmic number of points, and merging the resulting coresets yield a coreset for the entire data set. In other words, one needs to run the coreset procedure on only a small subset (or batch) of the input points and then use the standard merge-and-reduce procedure to combine the resulting coresets to form a coreset for the entire data set.
>
> The merge-and-reduce procedure is a traditional technique in coreset-construction dating back to the work of Har-Peled and Mazumdar (2004) (for a recent exposition of this technique, see: Braverman et al. 2016, as we cited in Sec. 4.2 in our submission) that exploits the fact that coresets are composable and reducible, as explained in our general response above. Moreover, the merged coreset can be further reduced by noting that an epsilon-coreset of a say, delta-coreset, is an (epsilon + delta)-coreset. Both the chicken and the egg phenomena and merge-and-reduce techniques are covered in detail in the related work we cited in the section (Braverman et al. 2016).
>
> In light of your feedback, we have modified the text to clarify the exposition of the chicken and the egg phenomena and the merge and reduce technique.
>
> 2) Thank you for bringing up this ambiguity in the reported runtime. We have should have highlighted that the running time of the algorithm is approximately linear if the merge-and-reduce procedure above is used and the sufficient conditions on the sum of sensitivities for the existence of small coresets mentioned in the analysis section hold. We have modified our paper accordingly and these changes are reflected in the revised version, namely in Sec. 4.2.
>
> We want to emphasize that our algorithm introduces a novel way to accelerate SVM training in streaming settings, where traditional SGD-based approaches to approximately-optimal SVM training (e.g., Pegasos) currently cannot handle. Therefore, comparing the *offline* performance of our algorithm (designed to operate in streaming settings) to SGD-based approaches (which cannot operate in streaming settings) may not be the most appropriate comparison.
>
> 3) We agree and mention in our original submission that our work builds on the framework for coreset construction introduced by Langberg et al. (2010) and generalized by Feldman et al., (2011). However, as these authors also note, the main challenge in using the coresets framework lies in establishing accurate upper bounds on the sensitivity of each point using analytical and algorithmic techniques. In fact, the novelty in the most of the recently published coresets papers lies in the introduction of novel upper bounds on the sensitivity (typically, via bicriteria approximations). In our paper, we not only provide accurate, data-dependent upper bounds on the sensitivity of each point, but also establish lower bounds on the sensitivity, which enables us to classify the set of problem instances for which our algorithm is most suited.

---

> ### Author Response · Authors · 2017-12-18
> **Updated Experimental Results**
>
> Thank you again for your consideration. We have updated our submission with a revised manuscript that includes additional comparisons in the streaming setting and evaluations against competitive algorithms, including Pegasos and Core Vector Machines (CVMs). Please feel free to refer to our latest general response and revision for additional details.

---

### Author Response · Authors · 2017-12-11
**General Response**

We thank all the reviewers for their useful suggestions and careful consideration of our paper.  Your feedback has raised several points we need to clarify prior to providing detailed answers. We understand that there is a range of expertise in this community and will improve our exposition to make sure the paper is broadly accessible to the ICLR community.

	Our submission proposes a coreset-based approach to speeding up SVM training by constructing compact representations of massive data sets. The key idea is that an SVM trained on coresets, i.e., weighted subsets of the original input points, generated by our algorithm is provably competitive with the SVM trained on the full data set. In contrast to SGD-based approaches, e.g., Pegasos, our approach extends to streaming data cases, where the input data set is so large that it may not be possible to store or process all the data at one time, as is common with Big Data applications and for dynamic datasets where samples are inserted/deleted. This new computational model for SVM is enabled by combining our coreset construction algorithm with the merge-and-reduce technique. The merge-and-reduce technique is over a decade old and is now a standard technique. We included references in the paper revision.

	Our algorithm requires knowledge of the optimal SVM solution in order to generate the coreset. This seemingly paradoxical construction is known as the chicken and the egg phenomenon, which commonly arises in coresets literature, and is resolved by the fact that the original algorithm is *not* intended to be run against the full data set, but rather small partitions of the data set. The merge-and-reduce procedure is a traditional technique in coreset construction dating back to the work of Har-Peled and Mazumdar (2004) (for a recent description of this technique, see: Braverman et al., (2016) as we cited in Sec. 4.2 in our submission) that exploits the fact that coresets are *composable*, i.e., if S_1 is an epsilon-coreset for data set P_1 and S_2 is an epsilon-coreset for data set P_2, then the union S_1 \cup S_2 is an epsilon-coreset for P_1 \cup P_2, and *reducible*, i.e., a delta-coreset of an epsilon-coreset is a ((1 + epsilon)*(1 + delta) - 1)-coreset.

	Thus, if the merge-and-reduce technique is used, our algorithm is only run on small subsets of the original data set. The results are then appropriately merged together, which implies that despite the super-linear runtime required to compute the optimal solution, the overall runtime is polylog(n) * d^3 * n, ignoring epsilon-error and delta (probability of failure) factors (details can be found in Sec. 4.2 of our revision). In our original submission, we show how this construction can be further sped up by using an efficient method to obtain a coarse, but near optimal solution, e.g., via Pegasos. We have included an additional lemma in the manuscript to clarify this point and to extend our prior analysis to this case, as requested by AnonReviewer3. We have also added details to sections 4.2 and 4.3 to further clarify the chicken and the egg phenomenon and the merge and reduce technique.

---

### Author Response · Authors · 2017-12-18
**Improved Results Section**

We wanted to update the reviewers and readers that our latest revision contains additional experimental results that evaluate and compare the performance of our algorithm with that of state-of-the-art. In particular, our latest revision contains the following additional experimental results:

1) Comparisons with Pegasos (Fig. 2)
2) Comparisons with uniform subsampling in the streaming setting where the data points arrive one-by-one (Fig. 5)
3) Comparisons with Core Vector Machines (CVMs) (Fig. 6).

Due to space constraints and our consideration that our theoretical results may have been of higher interest to the community, we were not able to fit all of these additional results in our original submission. However, in the case that our paper is accepted, we will certainly investigate ways to include these additional results in the final version of our paper.

---

> ### Comment · AnonReviewer1 · 2018-01-13
> **borderline**
>
> I checked the new experimental results.
>
> The new sampling method provides moderate improvement over the naive uniform sampling (in many cases).
> The running time part is not so convincing, as in many cases, it is significantly slower than other methods.
> Also, some text explaining those figures should be helpful.
>
> why the last figure in Figure 6 only has 2 curves?
>
> The new results are certainly helpful.
> But in my opinion, the paper may not be a clear accept of ICLR.

---

> > ### Author Response · Authors · 2018-01-13
> > **Response to AnonReviewer1**
> >
> > Thank you for the additional consideration.
> >
> > 1) Regarding the *offline* running time of our algorithm, we include below the response that we had posted earlier regarding the runtime comparisons. In short, our algorithm, unlike prior approaches, can be applied to streaming settings where it may not be possible to store or process all the data at one time, as is common with Big Data applications and for dynamic datasets where samples are inserted/deleted.
> >
> > Prior response regarding runtime:
> > ---
> > We want to emphasize that our algorithm introduces a novel way to accelerate SVM training in streaming settings, where traditional SGD-based approaches to approximately-optimal SVM training (e.g., Pegasos) currently cannot handle. Therefore, comparing the *offline* performance of our algorithm (designed to operate in streaming settings) to SGD-based approaches (which cannot operate in streaming settings) may not be the most appropriate comparison.
> > ---
> >
> > 2) The last graph in Fig. 6 actually contains all of the 4 curves, where uniform sampling, our coreset, and All Data essentially overlap (at either 0 or very close to 0 relative error).  This is due to CVM's poor performance on this particular data set combined with the good performance (i.e., relative error very close to 0) of both uniform sampling and our coreset. We will recreate the figure to reflect this overlap of curves more clearly and will also add further explanatory text as requested.

---

### Decision · Program_Chairs · 2018-01-29
**ICLR 2018 Conference Acceptance Decision**

**Decision:**

Reject

**Comment:**

While the paper shows some encouraging results for scaling up SVMs using coreset methods, it has fallen short of making a fully convincing case, particularly given the amount of intense interest in this topic back in the heydey of kernel methods. When it comes to scalability, it has become the norm now to benchmark results on far larger datasets using parallelism, specialized hardware in conjunction with algorithmic speedups (e.g., using random feature methods, low-rank approximations such as Nystrom and other approaches). As such the paper is unlikely to generate much interest in the ICLR community in its current form.